# OUT OF THE ORDINARY: SPECTRALLY ADAPTING REGRESSION FOR COVARIATE SHIFT

## ABSTRACT

Designing deep neural network classifiers that perform robustly on distributions differing from the available training data is an active area of machine learning research. However, out-of-distribution generalization for regression—the analogous problem for modeling continuous targets—remains relatively unexplored. To tackle this problem, we return to first principles and analyze how the closed-form solution for Ordinary Least Squares (OLS) regression is sensitive to covariate shift. We characterize the out-of-distribution risk of the OLS model in terms of the eigenspectrum decomposition of the source and target data. We then use this insight to propose a method for adapting the weights of the last layer of a pre-trained neural regression model to perform better on input data originating from a different distribution. We demonstrate how this lightweight spectral adaptation procedure can improve out-of-distribution performance for synthetic and real-world datasets.

## 1 INTRODUCTION

Despite their groundbreaking benchmark performance on many tasks—from image recognition and natural language understanding to disease detection (Balagopalan et al., 2020; Krizhevsky et al., 2017; Devlin et al., 2018)—deep neural networks (DNNs) tend to underperform when confronted with data that is dissimilar to their training data (Geirhos et al., 2020; D'Amour et al., 2020; Arjovsky et al., 2019; Koh et al., 2021). Understanding and addressing *distribution shift* is critical for the real-world deployment of machine learning (ML) systems. For instance, datasets from the WILDS benchmark (Koh et al., 2021) provide real-world case studies suggesting that poor performance at the subpopulation level can have dire consequences in crucial applications such as monitoring toxicity of online discussions, or tumor detection from medical images. Furthermore, DeGrave et al. (2021) demonstrated that models trained to detect COVID-19 from chest X-Rays performed worse when evaluated on data gathered from hospitals that were not represented in the training distribution. Unfortunately, poor out-of-distribution (OOD) generalization remains a key obstacle to broadly deploying ML models in a safe and reliable way.

While work towards remedying these OOD performance issues has been focused on classification, predicting continuous targets under distribution shift has received less attention. In this paper, we present a lightweight method for updating the weights of a pre-trained regression model (typically a neural network, in which case only the final layer is updated). This method is motivated by a theoretical analysis that yields a concrete reason, which we call *Spectral Inflation*, to explain why regressors may fail under covariate shift, a specific form of distribution shift. We then propose a post-processing method that improves the OOD performance of regression models in a synthetic experiment and three real-world datasets.

## 2 BACKGROUND

Distribution shift problems involve training on inputs $X$ and target labels $Y$ sampled from $P(X, Y)$, then evaluating the resulting model on a distinct distribution $Q(X, Y)$. Several learning frameworks consider different forms of distribution shift, depending on the structure of $P$ and the degree of prior knowledge about $Q$ that is available. For example in Domain Adaptation (DA) (Ben-David et al., 2006), unlabelled data (*unsupervised DA*) or a small number of labelled examples (*semi-supervised DA*) from $Q$ are used to adapt a model originally trained on samples from $P$. In some of our

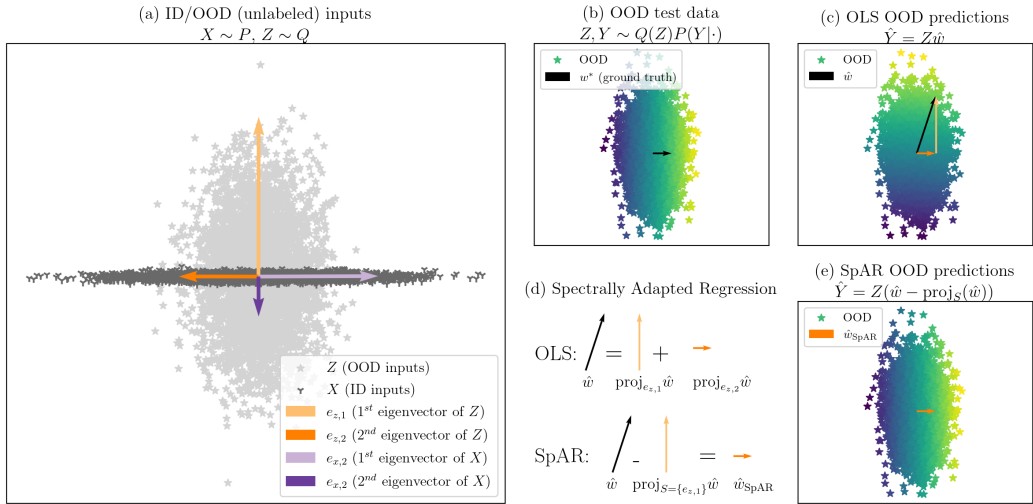

Figure 1: **Ordinary Least Squares Regression under Covariate Shift.** (a) Points are 2D input samples in the training set $X$ and test set $Z$. The in-distribution (ID) training data demonstrates nearly zero vertical variance, while the out-of-distribution (OOD) test data varies significantly in this direction. (b) Samples in $Z$ colored according to their true, noiseless labels $Zw^*$. (c) Samples in $Z$ colored according to their OLS predictions $Z\hat{w}$. Crucially, to minimize training risk, *OLS learns to weigh the vertical component highly* causing erroneous predictions OOD. (e) SpAR identifies a spectral subspace $S$ where train/test variance differ the most, and projects it out. Thus, the regressor created by SpAR ignores the direction with high variance and nearly recovers $w^*$.

experiments, we conduct unsupervised DA. In others the setting is very similar to unsupervised DA, with the exception that we update our model directly on the unlabeled test examples $X_{te} \sim Q(X)$ rather than on an independent sample $X' \sim Q(X)$ not used for evaluation. This setting is realistic and relevant to machine learning (Shocher et al., 2018; Sun et al., 2020; Bau et al., 2020).

We also assume the distribution shift is due to *covariate shift*, where the conditional distribution over the evaluation data $Q(Y|X)$ is equal to the conditional distribution over the training data $P(Y|X)$, but the input marginals $P(X)$ and $Q(X)$ differ. This broadly studied assumption (Sugiyama et al., 2007; Gretton et al., 2009; Ruan et al., 2021) states that the sample will have the same relationship to the label in both distributions. Within this setting, we turn our attention to the regression problem.

## 3    ROBUST REGRESSION BY SPECTRAL ADAPTATION

Least-squares regression has a known closed-form solution that minimizes the training loss, and yet this solution is not robust to covariate shift. In this section we show *why* this is the case by characterizing the OOD risk in terms of the eigenspectrum of the source and (distribution-shifted) target data. We then use insights from our theoretical analysis to derive a practical post-processing algorithm that uses unlabeled target data to adapt the weights of a regressor previously pre-trained on labeled source data. The adaptation is done in the spectral domain by first identifying subspaces of the target and source data that are misaligned, then projecting out the pre-trained regressor's components along these subspaces. We call our method **Spectral Adapted Regressor (SpAR)**.

### 3.1    ANALYZING OLS REGRESSION UNDER COVARIATE SHIFT

We begin with the standard Ordinary Least Squares (OLS) data generating process (Murphy, 2022). Rows of the input data matrix, $X \in \mathbb{R}^{N \times D}$, are i.i.d. samples from an unknown distribution $P$ over $\mathbb{R}^D$; these can be any representation, including one learned by a DNN from training samples. The rows of the evaluation input data, $Z \in \mathbb{R}^{M \times D}$, are generated using a different distribution $Q$ over $\mathbb{R}^D$. Analyzing final layer representations is useful as DNN architectures typically apply linear

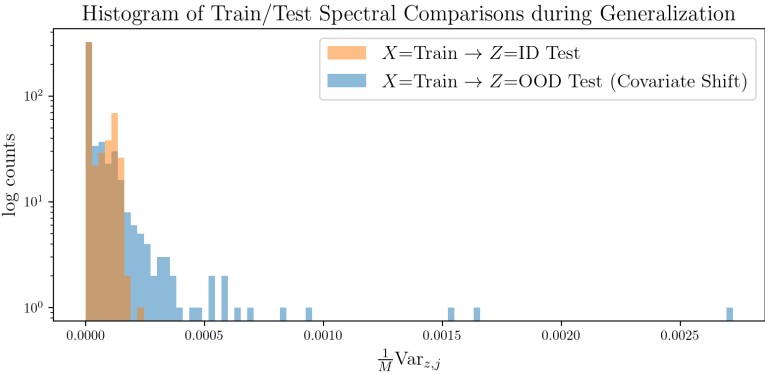

Figure 2: **Spectral Inflation.** We use the PovertyMap-WILDS dataset (Koh et al., 2021) to investigate how input spectra change when a regressor trained on real-world data generalizes to (perhaps shifted) test data. $X$ and $Z$ are composed of representations from a DNN. $Z$ represents data either from an in-distribution or out-of-distribution test set. $\text{Var}_{z,j}$, as defined in Equation 6, measures the amount of *Spectral Inflation*—small amounts of training set variation becoming large at test time—occurring along a given test eigenvector. Because each test sample has a different number of examples $M$, we normalize for a fair comparison. We see that when $Z$ is an out-of-distribution sample, much more spectral inflation occurs than when we generalize to an in-distribution sample.

models to these to make predictions. Targets depend on $X$ and $w^*$, a labeling vector in $\mathbb{R}^D$, and a noise term $\epsilon$. The targets associated with the test data $Z$ use the same true labeling vector $w^*$ but do not include a noise term as it introduces irreducible error:

$$X \sim P^N, \quad Y_X = Xw^* + \epsilon, \quad \epsilon \sim \mathcal{N}(0, \sigma^2 I), \quad Z \sim Q^M, \quad Y_Z = Zw^*. \tag{1}$$

The estimated regressor $\hat{w}$ that minimizes the expected squared error loss has the following form (Murphy, 2022), using $X^\dagger$, the Moore-Penrose Pseudoinverse of $X$, and its singular value decomposition, $X^\dagger = V_X D_X^\dagger U_X^\top$:

$$\arg\min_w \mathbb{E}[\|Y_X - Xw\|_2^2] = \hat{w} = X^\dagger Y_X = V_X D_X^\dagger U_X^\top Y_X. \tag{2}$$

We refer to $\hat{w}$ as the "OLS regressor" or "pseudoinverse solution". Our primary expression of interest will be the expected loss of $\hat{w}$ under covariate shift, which is the squared error between the true labels $Y_Z$ and the values predicted by our estimator $\hat{w}$. Specifically, we will analyze the expression:

$$\text{Risk}_{\text{OLS-OOD}}(\hat{w}) = \mathbb{E}[\|Y_Z - Z\hat{w}\|_2^2]. \tag{3}$$

In addition to using the singular value decomposition $X = U_X S_X V_X^\top$, we can also use the singular value decomposition of the target data $Z = U_Z S_Z V_Z^\top$. We define $\lambda_{x,i}, \lambda_{z,i}$ to be the $i^{th}$ singular values of $X$ and $Z$, respectively, and $e_{x,i}, e_{z,i}$ their corresponding unit-length right singular vectors. We will also refer to $\lambda_{x,i}^2, \lambda_{z,i}^2$ and $e_{x,i}, e_{z,i}$ as eigenvalues/eigenvectors, as they comprise the eigenspectrum of the uncentered covariance matrices $X^\top X$ and $Z^\top Z$. We use the operator $\text{Rows}()$ to represent the set containing the rows of a matrix. The OOD risk of $\hat{w}$ is presented in the following theorem in terms of interaction between the eigenspectra of $X$ and $Z$:

**Theorem 1** *Assuming the data generative procedure defined in Equations 1, and that $w^* \in \text{Span}(\text{Rows}(X))$ and $\text{Rows}(Z) \subset \text{Span}(\text{Rows}(X))$, the OOD squared error loss of the estimator $\hat{w} = X^\dagger Y$ is equal to:*

$$\mathbb{E}[\|Y_Z - Z\hat{w}\|_2^2] = \sigma^2 \sum_{i=1}^D \sum_{j=1}^D \frac{\lambda_{z,j}^2}{\lambda_{x,i}^2} \langle e_{x,i}, e_{z,j} \rangle^2 \mathbb{1}[\lambda_{x,i} > 0]. \tag{4}$$

This theorem indicates that if the samples in $Z$ present a large amount of variance along the vector $e_{z,j}$, resulting in a large eigenvalue $\lambda_{z,j}^2$, but the training set $X$ displays very little variance along

vectors at very similar angles, $\hat{w}$ will incur high loss. We refer to this scenario, when an eigenvector demonstrates this spike in variance at test time, as **Spectral Inflation**. An illustration of Spectral Inflation and its consequences are depicted in Figure 1, and we present evidence of Spectral Inflation occurring in DNN representations in a real-world dataset in Figure 2. The analysis follows from the cyclic property of the trace operator, which allows us to isolate the noise term $\epsilon$. This, in turn, enables a decomposition of the remaining expression in terms of the two eigenspectra of $Z^\top Z$ and $X^\top X$. A full derivation of this decomposition is available in Appendix B.

## 3.2 SPECTRAL ADAPTATION THROUGH PROJECTION

We now focus on identifying the eigenvectors occupying the rows of $V_Z^\top$ that contribute significantly to the expected loss described in Equation 4, and use them to construct a subset $S \subseteq \mathrm{Rows}(V_Z^\top)$. We then use $S$ to construct a new regressor $w_{\mathrm{proj}}$, by projecting $\hat{w}$ onto the subspace spanned by the eigenvectors in $S^c$, the complement of $S$:

$$w_{\mathrm{proj}} = \hat{w} - \sum_{e \in S} \langle \hat{w}, e \rangle e. \tag{5}$$

This regressor is not influenced by the Spectral Inflation displayed along each eigenvector in $S$, as $w_{\mathrm{proj}}$ exists in a subspace orthogonal to the subspace spanned by the vectors in $S$. Specifically, we can decompose the loss for this estimator $w_{\mathrm{proj}}$ into a sum over each eigenvector in $\mathrm{Rows}(V_Z^\top)$, where the contribution of the eigenvector $e_{z,j}$ to the loss is determined by whether that eigenvector is included in the set $S$. The following theorem expresses the expected OOD loss of $w_{\mathrm{proj}}$:

**Theorem 2** *Taking on the same assumptions as Theorem 1, the regressor $w_{\mathrm{proj}}$ constructed using a set $S \subseteq \mathrm{Rows}(V_Z^\top)$ as defined in Equation 5, has the following expected OOD squared error loss:*

$$\mathbb{E}[\|Y_Z - Z w_{\mathrm{proj}}\|_2^2] = \sum_{j, e_{z,j} \in S^c} \underbrace{\sigma^2 \sum_{i=1}^D \frac{\lambda_{z,j}^2}{\lambda_{x,i}^2} \langle e_{x,i}, e_{z,j} \rangle^2 \mathbb{1}[\lambda_{x,i} > 0]}_{\mathrm{Var}_{z,j}} + \sum_{j, e_{z,j} \in S} \underbrace{\langle w^*, e_{z,j} \rangle^2 \lambda_{z,j}^2}_{\mathrm{Bias}_{z,j}}.$$

The proof for this theorem is similar to the proof of Theorem 1 in that it uses the cyclic property of the trace to isolate the noise term. We then use the fact that each $e_{z,j} \in S$ is an eigenvector of $Z^\top Z$ to further decompose the expression. A full derivation of this decomposition is included in Appendix C. This case-like decomposition of the loss motivates our definition of the two different loss terms a single eigenvector $e_{z,j}$ can contribute to the overall expected loss. For a given eigenvector $e_{z,j}$ with associated eigenvalue $\lambda_{z,j}^2$, we will incur its **variance** loss if $e_{z,j} \notin S$, and its **bias** loss if $e_{z,j} \in S$, where the variance loss $\mathrm{Var}_{z,j}$ and bias loss $\mathrm{Bias}_{z,j}$ are defined as:

$$\mathrm{Bias}_{z,j} = \langle w^*, e_{z,j} \rangle^2 \lambda_{z,j}^2, \quad \mathrm{Var}_{z,j} = \sigma^2 \sum_{i=1}^D \frac{\lambda_{z,j}^2}{\lambda_{x,i}^2} \langle e_{x,i}, e_{z,j} \rangle^2 \mathbb{1}[\lambda_{x,i} > 0]. \tag{6}$$

$\mathrm{Var}_{z,j}$ is closely tied with the Spectral Inflation of an eigenvector, as $\mathrm{Var}_{z,j}$ will be large if $e_{z,j}$ demonstrates Spectral Inflation at test time. In this case if $e_{z,j} \notin S$, $w_{\mathrm{proj}}$ will have higher loss as a consequence of the label noise on the training examples distributed along this eigenvector. On the contrary, $\mathrm{Bias}_{z,j}$ is determined by the cosine similarity between the true labeling regressor $w^*$ and the eigenvector $e_{z,j}$. High cosine similarity means that this eigenvector makes a large contribution to determining a sample's label. If $e_{z,j} \in S$ and $e_{z,j}$ has a large cosine similarity to $w^*$, $w_{\mathrm{proj}}$ will incur a high amount of loss as it is orthogonal to this important direction.

## 3.3 PROJECTION REDUCES OUT-OF-DISTRIBUTION LOSS

Thus far, we have presented a decomposition for the expected loss of an estimator that is equal to the pseudoinverse solution $\hat{w}$ projected into the ortho-complement of the span of the set $S \subseteq \mathrm{Rows}(V_Z^\top)$. In this subsection, we present a means for constructing the set $S$ to minimize the expected loss by comparing $\mathrm{Var}_{z,j}$ and $\mathrm{Bias}_{z,j}$ for each test eigenvector $e_{z,j}$.

The ideal set $S^* \subseteq \mathrm{Rows}(V_Z^\top)$ would consist solely of the eigenvectors $e_{z,j}$ that have a greater variance loss than bias loss. Formally, this set would be constructed using the following expression:

$$S^* = \left\{ e_{z,j} : e_{z,j} \in \mathrm{Rows}(V_Z^\top), \mathrm{Var}_{z,j} \geq \mathrm{Bias}_{z,j} \right\}. \tag{7}$$

The following theorem demonstrates that using the set $S^*$ would give us a regressor that achieves superior OOD performance than the pseudoinverse solution.

**Theorem 3** *Under the same assumptions as Theorem 1, the regressor $w_{\mathrm{proj}}$ constructed as in Equation 5 using the set $S^*$ (cf. Equation 7) can only improve on the OOD squared error loss of the pseudoinverse solution $\hat{w}$:*

$$\mathbb{E}[\|Y_Z - Z\hat{w}\|_2^2] \geq \mathbb{E}[\|Y_Z - Zw_{\mathrm{proj}}\|_2^2]. \tag{8}$$

### 3.4 Eigenvector Selection Under Uncertainty

Theorem 3 shows that a regressor based on the set $S^*$ works better OOD. Finding $S^*$ would be easy if we knew both $\mathrm{Var}_{z,j}$ and $\mathrm{Bias}_{z,j}$ for each test eigenvector $e_{z,j}$. While we can calculate $\mathrm{Var}_{z,j}$ directly, $\mathrm{Bias}_{z,j}$ requires the true weight vector $w^*$, and so we can only *estimate* it using the pseudoinverse solution $\hat{w}$:

$$\widehat{\mathrm{Bias}}_{z,j} = \langle \hat{w}, e_{z,j} \rangle^2 \lambda_{z,j}^2 = (w^{*T} e_{z,j} + \epsilon^\top X^{\dagger\top} e_{z,j})^2 \lambda_{z,j}^2. \tag{9}$$

We fortunately have knowledge of some of the distributional properties of the dot product being squared: $\langle \hat{w}, e_{z,j} \rangle$. In particular, $w^{*\top} e_{z,j}$ is a fixed but unknown scalar and $\epsilon^\top X^{\dagger\top} e_{z,j}$ is the linear combination of several i.i.d. Gaussian variables with zero mean and variance $\sigma^2$.

$$\epsilon^\top X^{\dagger\top} e_{z,j} \lambda_{z,j} \sim \mathcal{N}(0, \mathrm{Var}_{z,j}), \quad \langle \hat{w}, e_{z,j} \rangle \lambda_{z,j} \sim \mathcal{N}(\sqrt{\mathrm{Bias}_{z,j}}, \mathrm{Var}_{z,j}). \tag{10}$$

The fact that $\widehat{\mathrm{Bias}}_{z,j}$ is a random variable makes it difficult to directly compare it with $\mathrm{Var}_{z,j}$. However, we can analyze the behavior of $\widehat{\mathrm{Bias}}_{z,j}$ when $\mathrm{Bias}_{z,j}$ is much larger than $\mathrm{Var}_{z,j}$, and vice versa, in order to devise a method for comparing these two quantities.

**(Case 1):** $\mathrm{Bias}_{z,j} \gg \mathrm{Var}_{z,j}$. In this case, $\mathrm{Bias}_{z,j} \approx \widehat{\mathrm{Bias}}_{z,j}$. This is because $w^{*\top} e_{z,j}$ will be much greater than $\epsilon^\top X^{\dagger\top} e_{z,j}$, which causes the former term to dominate in the RHS of Equation 9. Therefore $\widehat{\mathrm{Bias}}_{z,j} \gg \mathrm{Var}_{z,j}$.

**(Case 2):** $\mathrm{Var}_{z,j} \gg \mathrm{Bias}_{z,j}$. In this case, $\widehat{\mathrm{Bias}}_{z,j} \approx (\epsilon^\top X^{\dagger\top} e_{z,j})^2 \lambda_{z,j}^2$. This is because $w^{*\top} e_{z,j}$ will be much smaller than $\epsilon^\top X^{\dagger\top} e_{z,j}$, which causes the latter term to dominate in the RHS of Equation 9.

Therefore, since Equation 10 indicates $(\epsilon^\top X^{\dagger\top} e_{z,j}) \lambda_{z,j}$ is a scalar Gaussian random variable, we know the distribution of its square:

$$\widehat{\mathrm{Bias}}_{z,j} \sim \mathrm{Var}_{z,j} \times \chi_{df=1}^2, \tag{11}$$

where $\chi_{df=1}^2$ is a chi-squared random variable with one degree of freedom. If $\mathrm{CDF}_{\chi_{df=1}^2}^{-1}$ is the inverse CDF of the chi-squared random variable, then we have:

$$\Pr(\widehat{\mathrm{Bias}}_{z,j} \leq \mathrm{CDF}_{\chi_{df=1}^2}^{-1}(\alpha) \times \mathrm{Var}_{z,j}) = \alpha. \tag{12}$$

By applying these two cases, we can construct our set $S$ as follows:

$$S = \left\{ e_{z,j} : \widehat{\mathrm{Bias}}_{z,j} \leq \mathrm{CDF}_{\chi_{df=1}^2}^{-1}(\alpha) \times \mathrm{Var}_{z,j} \right\}. \tag{13}$$

The intuition behind this case-by-case analysis is formalized with the following proposition and lemma:

---

**Algorithm 1** Spectral Adapted Regressor (SpAR)

---

**Require:** Training Data $X, Y_X$, Unlabeled Test Distribution Data $Z$, Rejection Confidence $\alpha$

$\quad \hat{w} \leftarrow X^\dagger Y_X$

$\quad U_X, D_X, V_X^\top \leftarrow \text{SVD}(X)$

$\quad U_Z, D_Z, V_Z^\top \leftarrow \text{SVD}(Z)$

$\quad \hat{\sigma}^2 \leftarrow \text{MLE}(X, Y_X)$

$\quad S \leftarrow \{\}$                                                  $\triangleright$ Initialize the set S as empty

$\quad$ **for** $e_{z,j} \in \text{Rows}(V_Z^\top), \lambda_{z,j} \in \text{Diagonal}(D_Z)$ **do**    $\triangleright$ Iterate over Z's singular vectors and values

$\quad\quad\quad \text{Var}_{z,j} \leftarrow \hat{\sigma}^2 \sum_{i=1}^D \frac{\lambda_{z,j}^2}{\lambda_{x,i}^2} \langle e_{x,i}, e_{z,j} \rangle^2 \mathbb{1}[\lambda_{x,i} > 0]$

$\quad\quad\quad \text{Bias}_{z,j} \leftarrow \langle \hat{w}, e_{z,j} \rangle^2 \lambda_{z,j}^2$

$\quad\quad\quad$ **if** $(\text{CDF}_{\chi^2}^{-1}(\alpha) \times \text{Var}_{z,j}) \geq \text{Bias}_{z,j}$ **then**

$\quad\quad\quad\quad S \leftarrow S \cup \{e_{z_j}\}$            $\triangleright$ Include this vector in $S$ if its bias is below its variance threshold

$\quad\quad\quad$ **end if**

$\quad$ **end for**

$\quad w_{\text{proj}} \leftarrow \hat{w} - \sum_{e \in S} \langle \hat{w}, e \rangle e$                    $\triangleright$ Project out each of the selected vectors

$\quad$ **return** $w_{\text{proj}}$

---

**Proposition 1** *Making the same assumptions as Theorem 1, for a given choice of $\alpha \in [0, 1]$, the probability that test eigenvector $e_{z,j}$ is included in our set $S$ as defined in 13:*

$$\Pr(e_{z,j} \in S) = 1 - Q_{\frac{1}{2}}\left(\sqrt{\frac{\text{Bias}_{z,j}}{\text{Var}_{z,j}}}, \sqrt{\text{CDF}_{\chi^2_{df=1}}^{-1}(\alpha)}\right), \tag{14}$$

*where $Q_{\frac{1}{2}}$ is the Marcum Q-function with $M = \frac{1}{2}$.*

**Lemma 1** *Using the same assumptions as Proposition 1:*

$$\Pr(e_{z,j} \in S) \xrightarrow{\frac{\text{Bias}_{z,j}}{\text{Var}_{z,j}} \to \infty} 0, \qquad \Pr(e_{z,j} \in S) \xrightarrow{\frac{\text{Bias}_{z,j}}{\text{Var}_{z,j}} \to 0} \alpha. \tag{15}$$

Lemma 1 tells us that if we would incur significantly higher OOD loss from including $e_{z,j}$ in our set $S$ than excluding it, then $e_{z,j}$ **will not** be included in $S$. Similarly, if we would incur significantly higher OOD loss from excluding $e_{z,j}$ in our set $S$ than including it, then $e_{z,j}$ **will** be included in $S$.

Creating $w_{\text{proj}}$ in this way yields SpAR, a regressor tailored for a specific covariate shift (see Algorithm 1). Finally, this procedure requires the the variance of the training label noise, $\sigma^2$. We use a maximum likelihood estimate of this parameter (Murphy, 2022) from the training data.

## 4 EXPERIMENTS

In this section, we apply SpAR to a suite of real-world and synthetic datasets to demonstrate its efficacy and explain how this method overcomes some shortcomings of the pseudoinverse solution.

Here we use models that are optimized using gradient-based procedures. This contrasts with the main target of our analysis, the OLS solution (Equation 2), as $\hat{w}$ is not found using an iterative procedure. Despite these differences, our analysis remains relevant as the optimality conditions of minimizing the squared error loss ensure that gradient descent will converge to the OLS solution.

### 4.1 SYNTHETIC DATA

We establish a proof of concept by considering a synthetic data setting where we can carefully control the distribution shift under study. Specifically, we apply our approach to two-dimensional Gaussian data following the data generative process described in Section 3.1. Specifically, we sample our train and test data $X$ and $Z$ from origin-centered Gaussians with diagonal covariance matrices, where the variances of $X$ and $Z$ are $(5, 10^{-5})$ and $(1, 40)$ respectively.

Table 1: Mean and standard deviation of the squared error of our estimated regressors against various true labeling vectors. Each experiment setting is a different true weight vector (see Section 4.1).

| Synthetic Data | | | |
| --- | --- | --- | --- |
| Regression Method | Experiment 1 $(w_1^*)$ | Experiment 2 $(w_2^*)$ | Experiment 3 $(w_3^*)$ |
| ERM | 2.54e6 $\pm$ 3.84e6 | 2.54e6 $\pm$ 3.84e6 | 2.54e6 $\pm$ 3.84e6 |
| ERM + SPAR | **1.60e5** $\pm$ 3.13e3 | **2.82e0**$\pm$4.53 | **1.27e5** $\pm$ 2.51e3 |

We refer to the first and second indices of these vectors as the "horizontal" and "vertical" components and plot the vectors accordingly. The test distribution has much more variance along the vertical component in comparison to the training distribution. We experiment with three different true labeling vectors: $w_1^* = (.01, .99999995)^T$; $w_2^* = (0.9999995, 0.01)^T$; $w_3^* = (\frac{1}{\sqrt{5}}, \frac{2}{\sqrt{5}})^T$. The first two true labeling vectors represent functions that almost entirely depend on the vertical/horizontal component of the samples, respectively. $w_3^*$ depends on both directions, though it depends slightly more on the vertical component (cf. Figure 1). For each labeling vector, we randomly sample $Z, X$ and $\epsilon$ 10 times and calculate the squared error for both the OLS/Pseudoinverse solution $\hat{w} = X^\dagger Y_X$ (ERM) as well as $w_{\text{proj}}$, the regressor outputted by SpAR (ERM + SpAR). See Table 1 for results.

We first note that $\hat{w}$ is expected to have the same error regardless of the true labeling vector. Second, $w_{\text{proj}}$ outperforms $\hat{w}$ regardless of which true regressor is chosen. Our projection method is most effective when $w_2^*$ is being used to label the examples. This is due to the fact that it relies mostly on the horizontal component of the examples, which has a similar amount of variance at both train and test time. As a result, SpAR is able to project out the vertical component while retaining the bulk of the true labeling vector's information. An example showing why this projection method is useful when $w_2^*$ is being used to label the examples is depicted in Figure 1. Here, $\hat{w}$ significantly overestimates the influence of the vertical component on the samples' labels. SpAR is able to detect that it will not be able to effectively use the vertical component due to the large increase in variance as we move from train to test, and so it projects that component out of $\hat{w}$. Consequently, SpAR produces a labeling function nearly identical to the true labeling function.

## 4.2 Tabular Datasets

To test the efficacy of SpAR on real-world distribution shifts, we first experiment with two tabular datasets. Tabular data is common in real-world machine learning applications and benchmarks, particularly in the area of algorithmic fairness (Barocas et al., 2019). Therefore, it is important for any robust machine learning method to function well in this setting.

CommunitiesAndCrime, a popular dataset in fairness studies, provides a task where crime rates per capita must be predicted for different communities across the United States, with some states held out of the training data and used to form an OOD test set (Redmond & Baveja, 2009; Yao et al., 2022). Skillcraft defines a task where one predicts the latency, in milliseconds, between professional video game players perceiving an action and making their own action (Blair et al., 2013). An OOD test set is created by only including players from certain skill-based leagues in the train or test set.

We train neural networks with one hidden layer in the style of Yao et al. (2022). We benchmark two methods: the first is standard training (ERM), in which both the encoder and the regressor are trained in tandem to minimize the training objective using a gradient-based optimizer, in this case ADAM (Kingma & Ba, 2014). The other method we benchmark is C-Mixup (Yao et al., 2022), a data augmentation technique that generalizes the Mixup algorithm (Zhang et al., 2017) to a regression setting. For this method, the encoder and regressor are optimized to minimize the error on both the original samples and the synthetic examples produced by C-Mixup. Data-augmentation techniques such as C-Mixup can be used in tandem with other techniques for domain adaptation, such as SpAR, to achieve greater results than either of the techniques on their own. Our results substantiate this.

We use the hyperparameters reported by Yao et al. (2022) when training both ERM and C-Mixup. After training, we apply SpAR to create a new regressor using the representations produced by the ERM model (ERM + SpAR) or C-Mixup model (C-Mixup + SpAR). For SpAR, we explored a few settings of the hyperparameter $\alpha$ (see Appendix L for a discussion), and use a fixed value of $\alpha = 0.999$ in all the experiments presented here. These new regressors replace the learned regression weight in the last layer. We similarly benchmark the performance of the Pseudoinverse solution

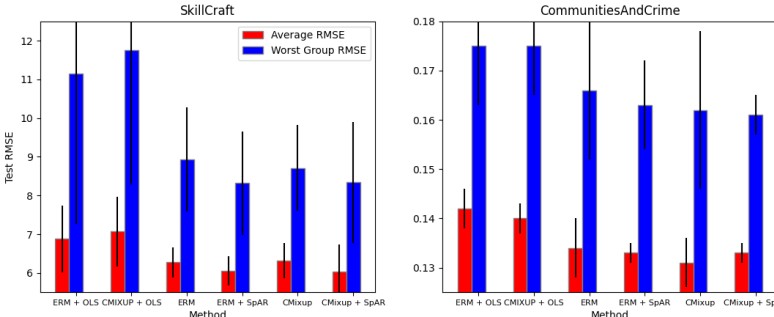

Figure 3: **Tabular data.** OOD RMSE for several methods, each averaged across 10 seeds.

by replacing the last layer weight with $\hat{w}$ (ERM/C-Mixup + OLS). Results from these tabular data experiments can be found in Figure 3. Exact numbers are presented in Table 5 in the Appendix.

Figure 3 shows that SpAR always produces a model with competitive or superior Average and Worst Group RMSE, regardless of the base model that it is applied to. We also experiment with tuning the hyperparameters for both the ERM and C-Mixup models in Appendix K. With no additional tuning for SpAR specifically, SpAR yields a model with the strongest worst-group performance.

## 4.3 POVERTYMAP - WILDS

We next examine the robustness of deep regression models under realistic distribution shifts in a high-dimensional setting. This experiment uses the PovertyMap-WILDS dataset (Koh et al., 2021), where the task is to regress local satellite images onto a continuous target label representing an asset wealth index for the region. PovertyMap provides an excellent test-bed for our method since, as seen in Figure 2, DNNs attempting to generalize OOD on this dataset suffer from Spectral Inflation.

Once again, for ERM and C-Mixup we use the hyperparameters suggested by Yao et al. (2022) and for SpAR we use $\alpha = 0.999$. When training these baselines, we follow Yao et al. (2022) and select the model checkpoint which best performed on a hold-out validation set as a form of early stopping. These choices help to create strong baselines. Results are presented in Table 2.

We can observe from Table 2 that applying SpAR can significantly improve worst-group performance while maintaining competitive average performance. As with Section 4.2, we further tune the hyperparameters for both the ERM and C-Mixup baselines in Appendix K. With **no tuning of SpAR specifically**, it is able to enhance the tuned baseline and yield the strongest worst-group performance. SpAR is also more computationally efficient than other robustness methods (see Appendix M).

Additionally, we experiment with an unsupervised domain adaptation setting where we used unlabeled target domain data distinct from the test set to perform adaptation with SpAR (Sagawa et al., 2021). We use the same base ERM and C-Mixup backbone models as presented in Table 2. We compare with many methods for robust ML, including some "in-processing" methods (Caron et al., 2020) which use the unlabelled data to define an additional objective that is optimized during training. Results are presented in Table 3. We find that even when using a sample distinct from the evaluation data, the use of SpAR on either ERM or C-Mixup yields the best performance. The worst group performance of C-Mixup + SpAR achieves state of the art performance on the PovertyMap-WILDS leaderboard for methods using unlabeled target domain data (Sagawa et al., 2021).

Table 2: **PovertyMap-WILDS.** Average OOD all-group and worst-group Spearman r across 5 splits.

| Method | $r_{all}(\uparrow)$ | $r_{wg}(\uparrow)$ |
|---|---|---|
| ERM | $0.793 \pm 0.040$ | $0.497 \pm 0.099$ |
| ERM + SpAR (Ours) | $\mathbf{0.794} \pm 0.046$ | $\mathbf{0.512} \pm 0.092$ |
| C-Mixup | $0.784 \pm 0.045$ | $0.489 \pm 0.045$ |
| C-Mixup + SpAR (Ours) | $\mathbf{0.794} \pm 0.043$ | $\mathbf{0.515} \pm 0.091$ |

Table 3: **PovertyMap-WILDS with unlabeled data.** In-processing methods and SpAR use unlabeled data that are distinct from the test set, but come from the same distribution (Sagawa et al., 2021).

| Robustness approach | Method | $r_{all}(\uparrow)$ | $r_{wg}(\uparrow)$ |
|---|---|---|---|
| — | ERM | $\mathbf{0.79} \pm 0.04$ | $0.50 \pm 0.10$ |
| Data augmentation | C-Mixup Yao et al. (2022) | $0.78 \pm 0.05$ | $0.49 \pm 0.05$ |
| (pre-processing) | Noisy Student Xie et al. (2020) | $0.76 \pm 0.08$ | $0.42 \pm 0.11$ |
| Self-supervised pre-training | SwAV Caron et al. (2020) | $0.78 \pm 0.06$ | $0.45 \pm 0.05$ |
| (pre-processing) | | | |
| Distribution alignment | DANN Ganin et al. (2016b) | $0.69 \pm 0.04$ | $0.33 \pm 0.10$ |
| (in-processing) | DeepCORAL Sun & Saenko (2016) | $0.74 \pm 0.05$ | $0.36 \pm 0.08$ |
| | AFN Xu et al. (2019) | $0.75 \pm 0.08$ | $0.39 \pm 0.08$ |
| Subspace alignment | RSD Chen et al. (2021) | $0.78 \pm 0.03$ | $0.44 \pm 0.09$ |
| (in-processing) | DARE-GRAM Nejjar et al. (2023) | $0.76 \pm 0.06$ | $0.44 \pm 0.05$ |
| Spectral adaptation | **ERM + SpAR** (Ours) | $\mathbf{0.79} \pm 0.04$ | $0.51 \pm 0.10$ |
| (post-processing) | **C-Mixup + SpAR** (Ours) | $\mathbf{0.79} \pm 0.04$ | $\mathbf{0.52} \pm 0.08$ |

## 5 RELATED WORK

Improving OOD performance is a critical and dynamic area of research. Our approach follows in the tradition of Transductive Learning (Gammerman et al., 1998) (adapting a model using unlabelled test data) and unsupervised Domain Adaptation (Ben-David et al., 2006; Farahani et al., 2021) (using distributional assumptions to model train/test differences, then adapting using unlabeled test inputs). Regularizing statistical moments between $P$ and $Q$ during training is a popular approach in unsupervised DA (Gretton et al., 2009) that has also been realized using deep neural networks (Ganin et al., 2016a; Sun et al., 2016). When transductive reasoning (adaptation to a test distribution) is not possible, additional structure in $P$—such as auxiliary labels indicating the "domain" or "group" that each training example belongs to—may be exploited to promote OOD generalization. Noteworthy approaches include Domain Generalization (Arjovsky et al., 2019; Gulrajani & Lopez-Paz, 2020) and Distributionally Robust Optimization (Hu et al., 2018; Sagawa et al., 2019; Levy et al., 2020).

Data augmentation is another promising avenue for improving OOD generalization (Hendrycks & Dietterich, 2019; Ovadia et al., 2019) that artificially increases the number and diversity of training set samples. The recently proposed C-Mixup method focuses on regression under covariate shift; it adapts the Mixup algorithm (Zhang et al., 2017) to regression by upweighting the convex combination of training examples whose target values are similar. This pre-processing approach complements our post-processing adaptation approach; in our experiments we find that applying SpAR to a C-Mixup model often yields the best results.

In this work we investigate covariate shift in a regression setting by analyzing how the distribution shift affects eigenspectra of the source/target data. We are not the first to study this problem, nor the first to use spectral properties in this investigation. Pathak et al. (2022) propose a new similarity measure between $P$ and $Q$ that can be used to bound the performance of non-parameteric regression methods under covariate shift. Wu et al. (2022) analyzes the sample efficiency of linear regression in terms of an eigendecomposition of the second moment matrix of individual data points drawn from $P$ and $Q$. Our work differs from these in that we go beyond an OOD theoretical analysis to propose a practical post-processing algorithm, which we find to be effective on real-world datasets.

## 6 CONCLUSION

This paper investigated the generalization properties of regression models when faced with covariate shift. In this setting, our analysis shows that the Ordinary Least Squares solution—which minimizes the training risk—can fail dramatically OOD. We attribute this sensitivity to *Spectral Inflation*, where spectral subspaces with small variation during training see increased variation upon evaluation. This motivates our adaptation method, SpAR, which uses unlabeled test data to estimate the subspaces with spectral inflation and project them away. We apply our method to the last layer of deep neural regressors and find that it improves OOD performance on several synthetic and real-world datasets. Our limitations include assumed access to unlabeled test data, and that the distribution shift in question is covariate shift. Future work should focus on applying spectral adaptation to other distribution shifts (such as concept shift and subpopulation shift) and to the domain generalization setting.

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
