# OpenReview forum: "Out of the Ordinary: Spectrally Adapting Regression for Covariate Shift"
_ICLR.cc/2024/Conference — Submitted to ICLR 2024_

### Official Review · Reviewer_FNwT · 2023-10-29

**Soundness:** 3 good
**Presentation:** 4 excellent
**Contribution:** 3 good
**Rating:** 8
**Confidence:** 4

**Summary:**

This paper tackles the problem of regression generalization to out of sample distributions. This is an important problem as many models suffer from poor performance when applied to different samples. The authors focus on the case of linear regression with fixed covariate shift and derive a clean decomposition of the out of sample error in terms of the model primitives. Through the decomposition the authors identify a cause of generalization error which they coin Spectral Inflation. Spectral Inflation occurs when the training set and evaluation set are misaligned in the sense that the dimensions along which most of the variation is explained do not coincide.

The paper proposes a novel post-processing algorithm that projects the OLS solution to a subspace that is well aligned with the evaluation set. The authors offer theoretical guarantees on how to choose the projection subspace given the data and researcher chosen hyper-parameters. Finally, they show the performance of the proposed method in a simulation exercise and across various empirical applications.

Overall the paper is very well written and I enjoyed reading it a lot.

**Strengths:**

* The paper tackles and important problem by considering a clean and simple setting. I found this very useful as it helps highlight the crux of the problem and offer an intuitive solution. Furthermore, the paper is well written, the exposition is good and the theoretical results are clean and technically correct.

* In terms of relevance, the decomposition results are not very surprising, but they do offer a straightforward and intuitive way to think of the generalization error in the context of linear regression. Researchers may find them useful as a framework for OOD error in these settings.

* The algorithm proposed is intuitive and the theory developed offers an interesting way of thinking why it is a good algorithm under different potential underlying models.

* The simulations and empirical examples are careful and offer a wide range of settings in which the method performs well.

**Weaknesses:**

* While the theoretical results are clean and correct, I wonder if the authors have solved the problem they set to solve rather than just offer one solution that is weakly better than OLS. I expand on this in the questions.

* While the proposed method performs well through the simulations and empirical tests it is unclear if it is better than other methods. For example, in Table 2 and 3 accounting for the errors it does not seem to be statistically different from the other methods (for example ERM in Table 2). The authors also do not compare it in simulations with alternative methods besides simple OLS. For instance, it seems that PCR might perform well in this setting.

**Questions:**

* In the decomposition results, what is the expectation being taken over? I thought that X and Z are treated as random, but in the proof in page 13 in the appendix it seems that X and Z are fixed. Are X and Z random or fixed? Are the decomposition results conditional on X and Z?

* Theorem 3 states that S^* is weakly better than the OLS solution, not that it is the best solution amongst all possible S. However in the paragraph above it is stated that it is the ideal set. Is it the case that S^* is the projection set that minimizes the expected loss amongst all possible S sets? It may be trivial, but it would be worth it fully explaining this in the main body of the text. If S^* is the the projection set that minimizes the expected loss then it should be stated as a theorem, if not then you should explain why you focus on S^* rather than the minimizing S.

* Can you use the plug in variance estimator to estimate the variance on the same data? Why is a sample split not necessary? (this could be trivial given the assumptions)

* How do you choose the hyperparameter alpha? Is there a data driven way to choose it or an optimal way of choosing it?

* It would be useful to decompose the MSE into bias and variance in the simulations to check that Spar indeed is trading off bias and variance as described by the theory in relation to OLS.

* What would change if Z was noisy? If we assume conditionally independent errors like for X, the conditional expectation would still be the same so it seems that most of the theory would go through with little changes (and the additional Z variance).

---

> ### Author Response · Authors · 2023-11-18
> **Response to Reviewer FNwT**
>
> We would first like to thank you for helping to improve our work. In our general response to all reviewers, we respond to some of your concerns about the size of the error bars on our Povertymap experiments, and how the source of this variance is not SpAR.
>
> **On the stochasticity of variables**: In our work, we only assume that the label noise on the labels Y is random. X and Z are assumed to be fixed matrices.
>
> **About the label noise variance estimator**: The estimator for the variance of the label noise can indeed be used without a data split. We use the estimator in Section 11.2.3.6 of Kevin Murphy’s Probabilistic Machine Learning (Murphy, Kevin P. Probabilistic machine learning: an introduction. MIT press, 2022). This is defined as simply the average of the residuals of the OLS solution applied to the training data.
>
> **Selection of $\alpha$ hyperparameter**: We thoroughly discuss the problem of choosing alpha in Section L of the appendix, as well as the sensitivity of performance to this hyperparameter.
>
> **About adding noise to the test labels**: If we were to add noise to the labels for Z that was independent of the noise on the X labels and had zero mean, in expectation this would simply introduce an irreducible loss term that was dependent on the variance of the Z label noise. Since this additive term is not influenced by any parameters we control (i.e. our choice of regressor), we omit it from our calculations for clarity’s sake. This is an important distinction to make, however, as some of the loss we observe in real world experiments could be due to label noise on the test examples.
>
> **On the strength of Theorem 3**: You are indeed correct that $S^*$ is the optimal set amongst all projection sets $S$ for decreasing the expected loss. This can be shown in a stronger version of theorem 3. While the projected regressor associated with $S^*$ having lower loss than the OLS solution is indeed a consequence of $S^*$ being optimal (take S equal to the empty set), it is perhaps the most important of these results as we are attempting to construct a regressor superior to OLS.

---

> > ### Comment · Reviewer_FNwT · 2023-11-20
> >
> > Thank you for your responses to my comments.
> >
> > Given your response to my question about Theorem 3, I think from a theoretical stand point it would be nice to either state the stronger version of the theorem in the text in Theorem form (and then have Theorem 3 as a corollary), or in the appendix and add more explanation in the main text. The issue with your current formulation is that there are other methods that also dominate OLS in this setting, so if you have a stronger theorem you should state it clearly.
> >
> > I am satisfied with your other answers, but I am not confident about whether your empirical applications show that your proposed method truly dominates ERM and, potentially, other methods. I am not completely familiar with the empirical benchmarks in the  literature, therefore if other referees agree with your responses regarding this I would be happy to increase my score to 8.

---

> > > ### Author Response · Authors · 2023-11-20
> > >
> > > Thank you for your response.
> > >
> > > Including the current form of Theorem 3 as a corollary of the proof that $S^*$ is optimal is an excellent idea. We will be sure to include this in the revisions to the draft.
> > >
> > > We welcome and encourage a discussion from all the reviewers on the empirical merits of our paper. We believe that SpAR's small but meaningful improvements upon baselines such as ERM are an important contribution for several reasons. First, SpAR is able to achieve state of the art performance on Povertymap, a challenging, real world benchmark for OOD regression. Second, SpAR improves upon ERM despite ERM being a very strong baseline (see the other methods presented in Table 3). Finally, SpAR is not only able to improve the performance of ERM, but it also improves the performance of other OOD robustness methods such as C-Mixup, leading to even larger performance gains in many cases.

---

> > > > ### Author Response · Authors · 2023-11-21
> > > >
> > > > As the discussion period nears the end, we wish to add a few comments that we hope will convince you that an increased score is merited. As you suggested, we welcome discussion from the other reviewers as well.
> > > >
> > > > Regarding the strength of our empirical findings, we wish to draw your attention to the additional benchmark datasets we have added (described above in our response to all reviewers). Since you mentioned deferring to other reviewers about the experiments, we also note that Reviewer TGia has raised their score in response to our rebuttal.
> > > >
> > > > Finally, we have followed your suggestion of benchmarking Principal Component Regression in our synthetic experiments. If we simply project the data onto the principal components (i.e. rotating the data) and then calculate the OLS solution, this performs similarly to OLS on the canonical basis. If we project the data onto only the leading principal component and then calculate the OLS solution, this yields performance similar to SpAR. This alludes to some important properties of SpAR's behaviour:
> > > >
> > > > - Behaving like PCR: In some circumstances, projecting onto the top-k principal components leads to lower loss. This is the case with our synthetic experiment. While PCR would require one to select how many principal components to project onto, SpAR automatically chooses which eigenvectors to project out.
> > > > - Behaving like OLS: If no Spectral Inflation takes place, then the regressor produced by SpAR is equal to the OLS solution. This is optimal when $bias_{z,j}$ is greater than $var_{z,j}$. In circumstances such as this, PCR will perform worse than OLS and therefore worse than SpAR.
> > > > - Unique behaviour: Since SpAR compares each test eigenvector's bias and variance separately, the eigenvectors it projects out can be interleaved throughout the spectrum instead of grouped together at the end of the spectrum. In this circumstance, SpAR will obtain lower expected loss than OLS or PCR. Analyzing the eigenvectors projected out by SpAR, we observe that SpAR does indeed project out a complex selection of eigenvectors, rather than just a top-k. In a revision to the work, we will include an example of which eigenvectors are projected out by SpAR.
> > > >
> > > > This demonstrates one of SpAR's most important properties: that it flexibly adapts to the covariate shift specified by the test data, rather than relying on the assumption that a certain adaptation will best perform OOD, as is the case with PCR. This is consistent with our other experiments where we find that adapting neural regressors using SPaR leads to consistent improvements across several real world datasets, both in the original submission and our newly added experiments from the discussion phase.

---

> > > > > ### Comment · Reviewer_FNwT · 2023-11-22
> > > > >
> > > > > I think the findings regarding the PCR vs SpAR simulations are intuitive and in some sense SpAR is an automatic method to do PCR, which means it should have this double-robustness property.
> > > > >
> > > > > After some consideration I have decided to raise my score to 8. Independently of the strength of the experimental evidence I think that the ideas proposed in the paper will be useful for future research on OOD.

---

> > > > > > ### Author Response · Authors · 2023-11-22
> > > > > > **thank you for your feedback**
> > > > > >
> > > > > > Thank you for your thoughtful consideration, and for you raising your score. We agree that adding PCR as a baseline in the synthetic study helps illuminate the adaptivity of SpAR. We believe the reader will benefit from this discussion and will include it in the camera ready version.

---

### Official Review · Reviewer_11sP · 2023-11-01

**Soundness:** 2 fair
**Presentation:** 3 good
**Contribution:** 2 fair
**Rating:** 5
**Confidence:** 3

**Summary:**

The authors study how deep regression models can be adapted to perform better under covariate shifts.

They do a detailed theoretical analysis of the ordinary least squares method and how it is affected by covariate shifts. Motivated by these findings, they propose a post-hoc method that can be used to update the final layer of pre-trained deep regression models, utilizing unlabeled data from the target distribution.

The proposed method is evaluated on three real-world regression datasets, two tabular datasets and one image-based. The regression performance is compared to that of standard training and C-Mixup (with or without the proposed final layer update).

**Strengths:**

I agree with the authors that out-of-distribution generalization specifically for _regression_ problems is relatively unexplored. Thus, the problem studied in this paper is definitely interesting and important.

The paper is well written overall, and the authors definitely seem knowledgeable.

Although I did not entirely follow all parts of the theoretical analysis in Section 3, I did find it quite interesting. Especially Figure 2. The resulting proposed method then also makes some intuitive sense overall.

**Weaknesses:**

I found it quite difficult to follow parts of Section 3, especially Section 3.4.

The experimental evaluation could be more extensive. The proposed method is applied just to three real-world datasets, of which two are tabular datasets where small networks with a single hidden layer are used.

The experimental results are not overly impressive/convincing. The gains of ERM+SpAR compared to the ERM baseline in Figure 3, Table 2 and Table 3 seem fairly small.

The computational cost of the proposed method (Algorithm 1) is not discussed in the main paper, and only briefly mentioned in the Appendix.

**Questions:**

1. Could the discussion of the computational cost be expanded and moved to the main paper? How does the cost of Algorithm 1 scale if X and/or Z contains a large number of examples? How about the memory requirements?

2. Could you evaluate the proposed method on at least on more image-based regression dataset? (one of the datasets in _"How Reliable is Your Regression Model’s Uncertainty Under Real-World Distribution Shifts?"_ (TMLR 2023) could perhaps be used, for example?)

3. The results in Table 3 seem odd, do all other baseline methods really degrade the regression performance compared to ERM?

4. Can you please discuss the results in Figure 3, Table 2 and Table 3 a bit more, the gains of ERM+SpAR compared to ERM seems quite small? Does the proposed method actually improve the performance of ERM in a significant way?


Minor things:
- Section 3.4, last paragraph: "the the variance" typo.
- I would consider modifying the appearance of Table 1 - 3, removing some horizontal lines.

---

> ### Author Response · Authors · 2023-11-18
> **Response to Reviewer 11sP**
>
> We would first like to thank you for helping to improve our work.  A number of your major concerns were shared with the other reviewers, and so please take a look at the general response to all reviewers. There, we respond to your concerns about the computational complexity of SpAR, how ERM presents a strong baseline, and how the variance in performance presented in Tables 2 and 3 is not due to SpAR but the multiple splits of the data that the models are being evaluated on.
>
> Thank you for your comments about the clarity of the writing. We will give the new concepts mentioned in Section 3.4 a more clear treatment in our revision of the work.

---

> > ### Author Response · Authors · 2023-11-21
> >
> > Reviewer 11sP, the discussion period ends on November 22nd. When you have a moment, please take a look at our general responses to all reviewers. You will find that we have included new experimental results using deep Resnet models. This includes an experiment using the ChairAngles-Tails dataset from the paper you suggested as a source for image-based benchmarks. Furthermore, we addressed your concerns about the strength of ERM as a baseline and the computational efficiency of SpAR. If our rebuttal has improved your opinion of the paper, please consider raising your score before the discussion period ends on November 22nd.

---

> > > ### Comment · Reviewer_11sP · 2023-11-22
> > > **Response to rebuttal**
> > >
> > > (Sorry for my late response. I have struggled to find enough time to both write responses as an author, and participate in the discussions as a reviewer)
> > >
> > > I have read the other reviews and all responses. Some of my concerns still remain.
> > >
> > > I thank the authors for the new experiments provided in the "Additional Results", this is a nice addition. However, I am not entirely convinced by these results. The improvement of ERM + SpAR compared to ERM is very small (0.6% and 0.5%). Given that the proposed method uses unlabeled data from the target distribution to update the final layer of the pre-trained ERM model, I think that more significant performance gains should be expected.
> > >
> > > Also, "How does the cost of Algorithm 1 scale if X and/or Z contains a large number of examples? How about the memory requirements?" from my first question is not really addressed.
> > >
> > > Thus, I am still borderline on this paper. It is not clear to me whether the proposed method actually improves the performance of ERM in a significant way. I would still need to see some more experimental results to be convinced, it would be interesting if you could evaluate the methods also on some of the real-world datasets from the "How Reliable is Your Regression Model’s Uncertainty Under Real-World Distribution Shifts?" paper (e.g., SkinLesionPixels or HistologyNucleiPixels).

---

> > > > ### Author Response · Authors · 2023-11-22
> > > >
> > > > # Reviewer 11sP
> > > > Thank you for considering our rebuttal and for your suggestions so far. We understand that you are still on the fence about whether to accept our submission, so we wish to address the points that you have raised so far during the discussion period.
> > > >
> > > > ## Computational complexity of SpAR
> > > > As we stated above, SpAR is cheaper than competing domain adaptation and data augmentation techniques. It relies on SVD decompositions of the source and target distribution, which is polynomial in the size of X or Z. The exact complexity depends on the implementation, but we believe the scipy package we used should be roughly equivalent to the MATLAB implementation, which is O(max(N,D) * min(N, D)^2) [https://www.mathworks.com/matlabcentral/answers/420057-what-is-the-complexity-of-matlab-s-implementation-of-svd] where X (or Z) is an N-by-D matrix. Because X (or Z) are neural network representations, before performing SVD we must first use the GPU to compute these activations in minibatches, aggregate the results together to get X (or Z). Aggregating the representations requires one network forward pass per N (although these are minibatched). These forward passes are much slower than computing the SVD, and we note that the forward passes would be required anyways to make predictions at test time. SpAR also benefits from the fact that no train-time regularizers need to be computed or backpropagated through, which makes it almost as fast as ERM (more specifically equal to ERM, plus the forward passes, then SVD). Even for the largest dataset we consider, SpAR is faster than *every* baseline we compare against besides ERM (See Table 10 in Appendix M).
> > > >
> > > > ## Choice of datasets
> > > > Thank you again for the pointer to Gustafsson et al 2023, which we agree is related and worth discussing. While their focus on uncertainty quantification differs from ours, the overall theme (that the reliability of neural regression models is relatively understudied and important in real world settings) aligns perfectly with the stated motivation for our method. Moreover, following your suggestion, we have incorporated one of these datasets, plus a relevant image regression benchmark from another paper, into our experiments section. The limiting factor in us adding additional benchmarks was time (speaking to your point about the lack of time during the ICLR discussion period). We did not try all eight benchmarks and pick the one that worked well. Rather, SpAR has worked well on each of the new datasets we have tried. We are happy to continue adding datasets in the camera ready version, although we believe that the evidence we have already collected (across 5 real-world datasets and 1 synthetic study) is sufficient to demonstrate the efficacy of our proposed method. We also note that Gustafsson et al 2023 was accepted to TLMR in the same week that we submitted this paper, so we only became aware of it through the discussion period here, which limited our ability to try all of of its benchmark datasets prior to submission.
> > > >
> > > > ## Performance of SpAR relative to baselines
> > > > We would argue that the main practical upsides of of SpAR are its adaptiveness and consistency. As noted by many other papers, OOD generalization without tuning hyperparameters on the target domain is very challenging, and standard training (ERM) remains a difficult baseline to beat. Because we choose which eigenspaces to project by measuring spectral inflation, this means that SpAR will default to OLS for distribution shifts where OLS is well suited. Likewise, if ERM would generalize better OOD, SpAR can match ERM's performance. However, for some distribution shifts, ERM may also suffer from some amount of spectral inflation. In these cases, SpAR can exceed ERM's performance (e.g. in the 4% boost in worst-group performance on PovertyMaps, which represents SOTA on this challenging dataset). So we can think of SpAR as a "best of all worlds" approach that we believe will interest practitioners. Notably, many of the competing methods we evaluated, especially domain adaptation approaches that (like SpAR) also use unlabeled target domain data, do much worse than SpAR or ERM.
> > > >
> > > > ## Non-empirical contributions of the paper
> > > > Aside from the experimental contributions, our paper provides theoretical contributions that we believe will help push the study of OOD-robust regression forward. Specifically, our analysis shows why OLS (the optimal regressor for the training distribution) fails OOD. We also characterize the OOD risk of regression under covariate shift, and introduce spectral inflation as way to measure this sensitivity.

---

> > > > > ### Comment · Reviewer_11sP · 2023-11-23
> > > > >
> > > > > Thank you for this additional detailed response. I do really appreciate your efforts to address my concerns.
> > > > >
> > > > >
> > > > > **"Computational complexity of SpAR":**
> > > > > This gives good additional details, thank you.
> > > > >
> > > > >
> > > > > **"Choice of datasets":**
> > > > > I appreciate the added experiments, and I definitely understand that time has been a limiting factor. I probably agree that the experimental evaluation now (with the two added datasets) is extensive enough, the problem is just that the results themselves are not particularly convincing. It is still not clear if the proposed method actually improves the performance of ERM in a significant way.
> > > > >
> > > > >
> > > > > **"Performance of SpAR relative to baselines":**
> > > > > It might be that I don't know the domain adaptation / generalization literature well enough, but it just does not seem surprising to me that the proposed method -- which uses unlabeled data from the target distribution to update the final layer of the pre-trained ERM model -- consistently gives small performance gains over ERM. In fact, this is the least I would expect from such a method to even be considered reasonable.
> > > > >
> > > > >
> > > > > **"Non-empirical contributions of the paper":**
> > > > > I do appreciate that the theoretical analysis can be valuable in itself, it is indeed an argument in favor of accepting the paper.
> > > > >
> > > > >
> > > > > In summary, I am still borderline on this paper. I might change my score to "6: marginally above", but I will need to take more time to carefully think about this during the reviewer-AC discussion period. Again, I want to thank the authors for their efforts to provide further clarifications. The problem studied in this paper is definitely interesting and important, and I do want to encourage more work in this direction.

---

### Official Review · Reviewer_TGia · 2023-11-01

**Soundness:** 2 fair
**Presentation:** 3 good
**Contribution:** 2 fair
**Rating:** 6
**Confidence:** 3

**Summary:**

In this paper, the authors introduce a novel post-processing technique to address the unsupervised domain adaptation challenge under the premise of covariate shift. This method stems from an intricate analysis of Ordinary Least Squares (OLS). The authors delve into the theoretical examination of the OLS loss in the context of covariate shift, leading to a proposal to project the estimator into a distinct subspace. The authors contend that selecting this subspace based on a comparative analysis of loss with respect to bias and variance across eigenvectors ensures enhanced performance on the target distribution. Empirical evaluations on multiple datasets substantiate the efficacy of the proposed approach.

**Strengths:**

1. The paper offers a rigor analysis of OLS in the presence of covariate shift. The proposed projection technique is sound, and the decomposition of the loss function is notably interesting.
2. The estimation strategy derived from finite samples is also sound.
3. The paper is generally well-written and easy to follow.

**Weaknesses:**

1. The paper sheds light on an interesting post-processing technique, aligning the linear layer from the source to the target domain. However, in the realm of deep learning, adapting the representational function across both domains is crucial. It raises the question: Can the proposed technique outperform other domain adaptation methods that also focus on refining the representation function? If such outperformance is challenging, is it feasible for the proposed post-processing technique to boost the performance of existing domain adaptation methods?
2. Stemming from the aforementioned concern, it would be enlightening to see the proposed method compared with a broader spectrum of domain adaptation baselines in Table 3 for the CommunitiesAndCrime and Skillcraft datasets. Moreover, an exploration of the combination of the proposed method and these baselines could be insightful.
3. For the PovertyMap-WILDS dataset, the setting aligns more with an out-of-distribution generalization task rather than traditional domain adaptation. Hence, it may be more judicious to include OOD methods for comparison. Furthermore, the performance enhancement attributed to the proposed method on this dataset seems marginal since the variance in performance exceeds the difference between the proposed method and the top-performing baseline.

**Questions:**

See the weakness part.

---

> ### Author Response · Authors · 2023-11-18
> **Response to Reviewer TGia**
>
> We would first like to thank you for helping to improve our work. Please see our general response to all reviewers, as we address your concerns about the computational complexity of SpAR.
>
> **Regarding representation-based methods for OOD generalization**: We present several methods which either implicitly or explicitly regularize the model’s representation as baselines in our experiments. In Table 3, both DANN and Deep-CORAL directly operate on the representation, while other methods such as C-Mixup and SwAV influence the entire network and therefore influence the representation as well. We note that SpAR is able to outperform all of these methods, achieving state-of-the-art performance on the Povertymap-WILDS dataset. It is also able to boost the performance of another method, C-Mixup.
>
> Furthermore, to your point about the importance of aligning representations across the training and testing distributions, SpAR can equivalently be thought of as a method that projects the test representations into a subspace spanned by the eigenvectors that do not demonstrate Spectral Inflation. This choice of subspace is informed by our theoretical work, and is the crux of our method’s effectiveness.

---

> > ### Comment · Reviewer_TGia · 2023-11-21
> >
> > I thank the authors for the response. I would like to raise my score to 6.

---

### Official Review · Reviewer_ZgCX · 2023-11-01

**Soundness:** 3 good
**Presentation:** 3 good
**Contribution:** 3 good
**Rating:** 6
**Confidence:** 4

**Summary:**

This paper explores the problem of out-of-distribution generalization for regression models. The authors first analyze the sensitivity of the Ordinary Least Squares (OLS) regression model to covariate shift and characterize its out-of-distribution risk. Then they use this analysis to propose a lightweight spectral adaptation procedure(called Spectral Adapted Regressor) for the last layer of a pre-trained neural regression model. The paper demonstrates the effectiveness of this method on synthetic and real-world datasets, and it works well with data enhancement techniques such as C-Mixup.

**Strengths:**

1. Although there has been extensive research on distribution shifts in classification tasks, the authors focus on regression models which is a relatively unexplored problem of out-of-distribution generalization. The authors provide a novel analysis of the sensitivity of the Ordinary Least Squares (OLS) regression model to covariate shift. And they propose a spectral adaptation procedure specifically tailored for regression models, this adds to the originality of the paper.
2. The authors provide a thorough analysis of the OLS regression model's out-of-distribution risk, utilizing the spectrum decomposition of the source and target data.
3.  The paper provides a concise abstract that outlines the problem, methodology, and results.

**Weaknesses:**

1. The proposed method assumes access to unlabeled test data for estimating the subspaces with spectral inflation. However, in practical scenarios, obtaining unlabeled test data may not always be feasible. It would be beneficial to explore alternative approaches or modifications to the method that do not rely on unlabeled test data.
2. The compared methods are limited to me. The persuasiveness of the proposed approach would be stronger if more comparative data could be provided.

**Questions:**

How computationally efficient is SpAR? The paper does not provide a detailed analysis of the computational efficiency of the proposed method. Considering the increasing complexity and size of neural regression models, it is important to assess the computational cost of the spectral adaptation procedure.

---

> ### Author Response · Authors · 2023-11-18
> **Response to Reviewer ZgCX**
>
> We would first like to thank you for helping to improve our work. Please see our general response to all reviewers, as we address your request for additional comparisons.
>
> **On the use of unlabelled data**: Investigating methods that make use of unlabelled data is an important endeavor for producing machine learning models that best make use of all of the resources available to them. In our work, we flexibly assume access to either the unlabelled evaluation data itself or a separate unlabelled data from the same distribution as the evaluation data. These are the assumed settings in the vast majority of research on OOD generalization (see the discussion in Section 5 of the paper). Consequently, positing an effective, efficient method that works in this regime is relevant to many machine learning researchers and practitioners alike.

---

> > ### Author Response · Authors · 2023-11-21
> >
> > Reviewer ZgCX, the discussion period ends on November 22nd. When you have a moment, please take a look at our general responses to all reviewers to find the additional comparisons that you requested in your review, as well as a discussion about SpAR’s computational efficiency. If our rebuttal has improved your opinion of the paper, please consider raising your score before the discussion period ends on November 22nd.

---

### Author Response · Authors · 2023-11-18
**General Response to all Reviewers**

We first thank all of the reviewers for their thoughtful and helpful comments. We would like to address some issues that several reviewers brought up in their reviews.

**Regarding the variance of Povertymaps-WILDS performance**: Evaluation on the Povertymap-WILDS dataset involves training and testing on _five distinct ID-OOD splits of the data_. Each of these splits require the model to generalize to a set of countries that the model did not observe during training. The difficulty of this task depends on the similarity of the countries in the training set and the test set, and so the difficulty of performing well varies across these five splits of the data. Therefore, since the results presented in Tables 2 and 3 are the average of the performance across these five splits, the error bars presented are an artifact of the varying difficulty across the five distribution shifts. Despite this, **ERM+SpAR is able to outperform ERM on 4/5 of the splits**, demonstrating that SpAR can consistently improve performance.

**About the strength of ERM as a baseline**: While it is counterintuitive that standard training (ERM) would outperform methods specifically designed to perform well in the face of distribution shift, this result is well documented [1,2]. In our work, we present very strong ERM baselines which are able to outperform many other methods for training robust models. We hypothesize that the significant improvement of ERM over OLS derives from the mitigation of spectral inflation due to the standard regularization in ERM training. Despite this, SpAR is able to yield models with superior performance to all other benchmarks, including ERM. On PovertyMap-WILDS, **SpAR’s performance is state of the art**.

**On SpAR’s computational efficiency**: For specific information about SpAR’s efficiency, we refer to Section M of the Appendix. After standard inference is used to collect the representations on the train and test sets, SpAR only requires performing SVD on these matrices. Since SpAR is a post-processing method, this is a one-time cost. This is a stark contrast from several other adaptation methods that require an expensive regularizer to be computed every epoch. For instance, training an ERM model from scratch and applying SpAR is four times faster than training with C-Mixup.


**Lack of distribution-shift regression benchmark datasets**: Our experiments consider two tabular benchmark datasets and one benchmark image-based dataset, PovertyMap.  This is due to the dearth of distribution-shift regression benchmark datasets. We thank reviewer 11sP for pointing us to the recent paper (published November 2023): "_How Reliable is Your Regression Model’s Uncertainty Under Real-World Distribution Shifts?_". In it the authors study uncertainty estimation, and propose several new distribution-shift regression benchmark datasets. They state: "While Koh et al. (2021) propose an extensive benchmark with various real-world distribution shifts, it only contains a single regression dataset…Ovadia et al. (2019) perform a comprehensive evaluation of uncertainty estimation methods under distribution shifts, but only consider classification tasks." We are investigating their proposed datasets and hope to have results to report before the end of the discussion period.


References:

[1] Sagawa, Shiori, et al. "Extending the WILDS benchmark for unsupervised adaptation." arXiv preprint arXiv:2112.05090 (2021),

[2] Gulrajani, Ishaan, and David Lopez-Paz. "In search of lost domain generalization." arXiv preprint arXiv:2007.01434 (2020)

---

### Author Response · Authors · 2023-11-21
**Additional Results**

Following suggestions from Reviewers 11sP, ZgCX, and TGia, we present additional results on the RCF MNIST dataset used in the work of Yao et al. [1], as well as the ChairAngles-Tails dataset presented in the work referenced by Reviewer 11sP [2]. Both of these are image datasets, and we use large Resnet models (Resnet 18 and Resnet 34) as our architecture of choice. We benchmark four baseline methods on these datasets: ERM, C-Mixup, DANN [3], and Deep CORAL [4]. After performing a hyperparameter sweep, we average results across 10 random seeds. In the tables below, we present both the baseline method's performance without any adaptation, as well as the performance once SpAR has been applied.

We observe results similar to what was presented in the original manuscript: SpAR is regularly able to improve performance across a wide variety of architectures, tasks, and training methods. On RCF MNIST, SpAR makes improvements on ERM (0.6% improvement), C-Mixup (1.2 %improvement) , Deep CORAL (2% improvement), and DANN (4% improvement). On ChairAngles, Spar makes improvements on ERM (0.5% improvement), C-Mixup (0.8% improvement) , Deep CORAL (2.3% improvement), and DANN (1.6% improvement).

The best performing baseline method varies across these two datasets, with ERM performing best on RCF MNIST and Deep CORAL performing best on ChairAngles-Tails. Despite these inconsistencies in baseline performance, SpAR consistently improves the performance of each method. This demonstrates SpAR's utility as a lightweight, efficient post processing method with a strong theoretical foundation that can be applied to a wide array of learned representations.

**RCF MNIST**

| Method | Baseline Average RMSE | Baseline + SpAR Average RMSE | Percent Improvement |

| ERM | **0.155** +- 0.006 | **0.154** +- 0.006 | 0.6% |

| C-Mixup | 0.158 +- 0.011 | 0.156 +- 0.009 | 1.2% |

| Deep CORAL | 0.167 +- 0.012 | 0.165 +- 0.010 | 2% |

| DANN | 0.177 +- 0.019 | 0.170 +- 0.015 | 4% |

**ChairAngles-Tails**

| Method | Baseline Average RMSE | Baseline + SpAR Average RMSE | Percent Improvement |

| ERM | 6.788 +- 0.634 | 6.753 +- 0.648 | 0.5% |

| C-Mixup | 6.504 +- 0.324 | 6.449 +- 0.325 | 0.8% |

| Deep CORAL | **5.978** +- 0.243 | **5.839** +- 0.259 | 2.3% |

| DANN | 6.440 +- 0.602 | 6.337 +- 0.603 | 1.6% |



References:

[1] Yao, Huaxiu, et al. "C-mixup: Improving generalization in regression." Advances in Neural Information Processing Systems 35 (2022): 3361-3376.

[2] Gustafsson, Fredrik K., Martin Danelljan, and Thomas B. Schön. "How Reliable is Your Regression Model's Uncertainty Under Real-World Distribution Shifts?." arXiv preprint arXiv:2302.03679 (2023).

[3] Ganin, Yaroslav, et al. "Domain-adversarial training of neural networks." The journal of machine learning research 17.1 (2016): 2096-2030.

[4] Sun, Baochen, and Kate Saenko. "Deep coral: Correlation alignment for deep domain adaptation." Computer Vision–ECCV 2016 Workshops: Amsterdam, The Netherlands, October 8-10 and 15-16, 2016, Proceedings, Part III 14. Springer International Publishing, 2016.

---

### Meta-Review · Area_Chair_NPN3 · 2023-12-07

**Metareview:**

This paper proposes a spectral projection method for the covariate shift regression. The main idea is to apply a spectral adaptation for the parameters of the last layer of neural networks after it is trained on the source data, based on a comparative analysis of losses with respect to the bias and variance across eigenvectors between the source and target domains. The method is validated in both synthetic covariate shift data and large-scale domain adaptation data in WILDS.

Strengths: all reviewers agree that the idea is interesting, the problem is important, the technique is sound and the presentation is relatively clear.

Weaknesses: the major concern is the experimental validation of the methods. The improvement over ERM and other domain adaptation methods seems to be limited.

As a post-processing method, it can be the case the influence on the model is limited. To further improve the method, it is worthwhile exploring combining this idea with the previous domain adaptation methods that focus on learning a generalizable representation from the source domain. I also recommend investigating the effect of the method on the model uncertainty.

**Justification For Why Not Higher Score:**

Even though the reviews are leaning towards positive, all reviewers, even the most positive ones point out the weakness in the empirical results. I take it as a serious problem and recommend another round of improvement.

**Justification For Why Not Lower Score:**

N/A

---

### Decision · Program_Chairs · 2024-01-16

Reject